# Sickle cell disease in Cameroon: Taking out the "neglect" and highlighting key opportunities for sustainable control

Yauba Saidu[1]*, Makia Christine Masong[2], Nwabufo Francoise[3], Budzi Michael Ngenge[1], Elvis Ndansi[4], Munoh Kenne Foma[5]

1 Clinton Health Access Initiative, Inc., Yaoundé, Cameroon, 2 Department of Social Sciences and Management, Catholic University of Central Africa, Yaoundé, Cameroon, 3 Family Health and Development Foundation, Yaoundé, Cameroon, 4 Unite for Health, Atlantic City, New Jersey, United States of America, 5 Presbyterian Healthcare Services, Hematology and Oncology, Albuquerque, New Mexico, United States of America

* ysaidu@clintonhealthaccess.org, yaubasaidu2011@gmail.com

**Data Availability Statement:** Anonymised data used for this research can be accessed from Figshare using this link: https://doi.org/10.6084/m9.figshare.26520400.v1.

## Abstract

Sickle Cell Disease (SCD) is a serious genetic disorder with astounding regional differences in childhood survival. Alarmingly, over 90% of children with SCD in SSA die before their fifth birthday. In Cameroon, approximately 7,000 children are born with SCD annually; however, most of them go undiagnosed until their fourth birthday, resulting into untold pain and suffering. Despite this, little is known about the barriers to optimal care and treatment for SCD in Cameroon. Here, we assess these barriers, and consider opportunities that could be leveraged, for a sustainable control of SCD in Cameroon. We conducted a qualitative study, with documentary analysis of key national and international policy documents related to SCD management in Cameroon; semi-structured interviews; and focus group discussions, used for data collection. Key informants were selected purposively, and met at the central level of the health system (in Yaoundé) and the operational level (Mfou health district). These were policy makers, health workers, parents with SCD children and teenagers with SCD. Several critical gaps exist which hinder SCD control in Cameroon. These include lack of a national sickle cell disease strategy, no proportional allocation of funds for SCD in the national budget, and gaps in service delivery. These are translated into healthcare providers having little knowledge on SCD, absence of SCD-specific indicators in the health information system, challenges accessing essential medicines, and limited awareness raising in communities on SCD. Still, several opportunities exist which could be leveraged for improving SCD care in Cameroon. These include the possibility of integrating SCD screening and care into well-established primary healthcare services like vaccination, antenatal care, and non-communicable disease clinics. In the light of such limited resource settings, considering opportunities for integration in existing health programs could go a long way to reduce morbidities and mortalities from SCD over the coming years.

**Funding:** This research was supported in part by grant number GV673603927 from Open Philanthropy. The funding was award to the Clinton Health Access Initiative, with Dr. Yauba Saidu, being the grant owner. The funders had no role in study design, data collection and analysis, decision to publish, or preparation of the manuscript.

**Competing interests:** The authors have declared that no competing interests exist.

## Introduction

Sickle Cell Disease (SCD) is a serious inherited disorder of red blood cells, which ranks as the most prevalent genetic disease globally [1]. Each year, over half a million children are born with the disease, with over 75% of these births occurring in sub-Saharan Africa (SSA) [2]. Regrettably, over 90% of children with this condition in SSA do not live to see their fifth birthday [2, 3]. Even those who surpass this crucial age face diminished prospects of reaching adulthood. This picture contrasts that in high-income countries, such as the United States of America and the United Kingdom, where mortality from SCD has seen a commendable reduction [4]. Indeed, in these settings most children born with the disease have been reported to live up to their 70th birthday, highlighting gross disparities in SCD care and treatment across the world.

The strides made in dramatically reducing mortality from SCD in high-income countries have partly been attributed to a combination of factors [4]. These include, identifying the sickle cell trait (SCT) in both parents during prenuptial and prenatal periods, and for the baby during post-natal period via newborn screening. The sickle cell trait (SCT) occurs when a person carries a copy of the sickle cell hemoglobin along with a normal hemoglobin, and unlike people with SCD, they do not usually have any of the SCD symptoms, though they can pass the trait unto their children. Early screening is foundational in preventing the birth of children with SCD, mitigating infant mortality, and initiating clinical interventions early enough to forestall complications arising from SCD later in life. However, these practical control strategies remain very limited in many SSA countries [4]. As a result, many children harboring the trait are left to endure a fate of remaining undiagnosed or untreated until they succumb to the disease. In parallel, these individuals and their families not only lack access to adequate care, but also bear the burden of stigmatization [5, 6], especially in settings like Nigeria [7], and Cameroon [8].

Cameroon has one of the highest prevalence of SCD in SSA, occupying the 6th position among countries with high burdens of SCD [9]. Approximately 30% of the country's estimated 30 million people carry the sickle cell trait (HbAS), and each year, approximately 7,000 children are born with the disease [8, 10]. Various reports have estimated the prevalence of SCD (precisely HbSS, the most severe form of SCD) in the general population to hover around 0.6–2% of the population [10]. Despite this high prevalence, many children with SCD in Cameroon are only diagnosed after their 4th birthday [11]. This reality which significantly deviates from the recommended screening at birth [4], suggests most children still elude early diagnosis. As a result, they face the grim prospect of under-5 mortality, and regrettably, most of this goes unrecorded. This failure to capture these children at birth ultimately contributes to several years lost to unmanaged pain and sickness, retarded growth, and missed school days among children suffering from the disease.

Despite these alarming challenges, little is known about the barriers to screening and the care and treatment of patients suffering from SCD in Cameroon [12]. This knowledge gap highlights the need to assess the current policy and funding landscapes, the organization for service delivery for SCD, and the perspectives of individuals with SCD in Cameroon. In this paper, we present the findings from a qualitative study that explored these barriers and propose measures that could be deployed, or opportunities which could be leveraged, to sustainably improve on various aspects of SCD care and treatment in Cameroon.

## Materials and methods

### Study design and setting

We conducted a cross-sectional study from May 1st, 2022 to 25th February 2024, which leveraged mixed qualitative methods: semi-structured interviews, focus group discussions, and

documentary analysis [13]. Interview guides were structured around main themes such as existing national control measures, access to health care for people living with SCD, community support, perceptions and individual lived experiences. The documentary analysis was focused mostly on existing policies and infrastructure guiding SCD control in Cameroon for key context information.

The study was conducted in Cameroon, a country stretching from the Gulf of Guinea to the lake Chad Basin, bordered by 6 countries: Nigeria (to the west), Chad (to the Northeast), Central African Republic (to the East), and Gabon, Congo, and Equatorial Guinea to the south. Cameroon has a total population of roughly 30 million inhabitants, with approximately 30% of them carrying the sickle cell trait (SCT).

The respondents from our study came from two levels: the central and the operational levels. At the central level, data were obtained from Policy Makers (PM) as well as selected staff at tertiary health facilities with experience in managing SCD patients. At the operational level, data were obtained from healthcare workers, parents of children living with SCD and teenagers with SCD in 5 rural communities in Mfou Health District. The Mfou Health District was of interest because of its hyperendemicity for malaria [14, 15], which suggests a possible high prevalence for SCT/SCD, based on reports which show a similarity of the geographical location of the two diseases [16, 17]. Moreover, the district has a relatively high under 5 years mortality [18], mostly stemming from severe anemia (hemoglobin level <8.0g/dl) [14]. In addition, informal narratives of "sick children" have been commonplace for decades within this area. Despite these, there is a paucity of data on SCT or SCD from this health district.

## Data collection procedure

Fig 1 depicts the procedure used to collect data for this study. From April 2022 to February 2023, a documentary analysis was used to provide context and develop upon information collected through triangulation for credibility. For the Documentary analysis (DA), physical documents were collected on site, and electronic documents were collected through conducted searches on PubMed, local news, and related blogs/reports (see S1 Table). This enabled us to identify the resources listed in Fig 1. Following this, 12 in-depth Interviews (IDI), and 3 Focus group discussions (FGD) involving policy makers (PM), healthcare workers (HW), parents of children with SCD and teenagers with SCD in the Mfou HD, were carried out. The documentary analysis was conducted in tandem with the group discussions and interviews (Fig 1). All key informants were identified purposively, based either on the fact they were living with SCD and resident in the Mfou health district, or were health workers within the Mfou district hospital and health centers, or involved in SCD control as policy makers or experienced SCD practitioners. Teenagers with SCD and their caregivers were identified either through health workers at the hospital/health centers, or met at organized SCD community group meetings.

## Ethical clearance and informed consent

An ethical clearance was accorded for parts of this research by the National Ethical Committee for Human Research in Cameroon (CE N˚ 0464/CRERSHC/2022). At the start of each interview or focus group discussion, an information notice was shared, and written informed consent was gotten from all participants above 18. For SCD patients younger than 18, written informed consent was gotten from parents or guardians, as well as their own assent.

## Analysis

Interviews and group discussions from recorded audios were transcribed in verbatim. During the documentary analysis, firstly documents were reviewed to assess an appraisal theme. Once

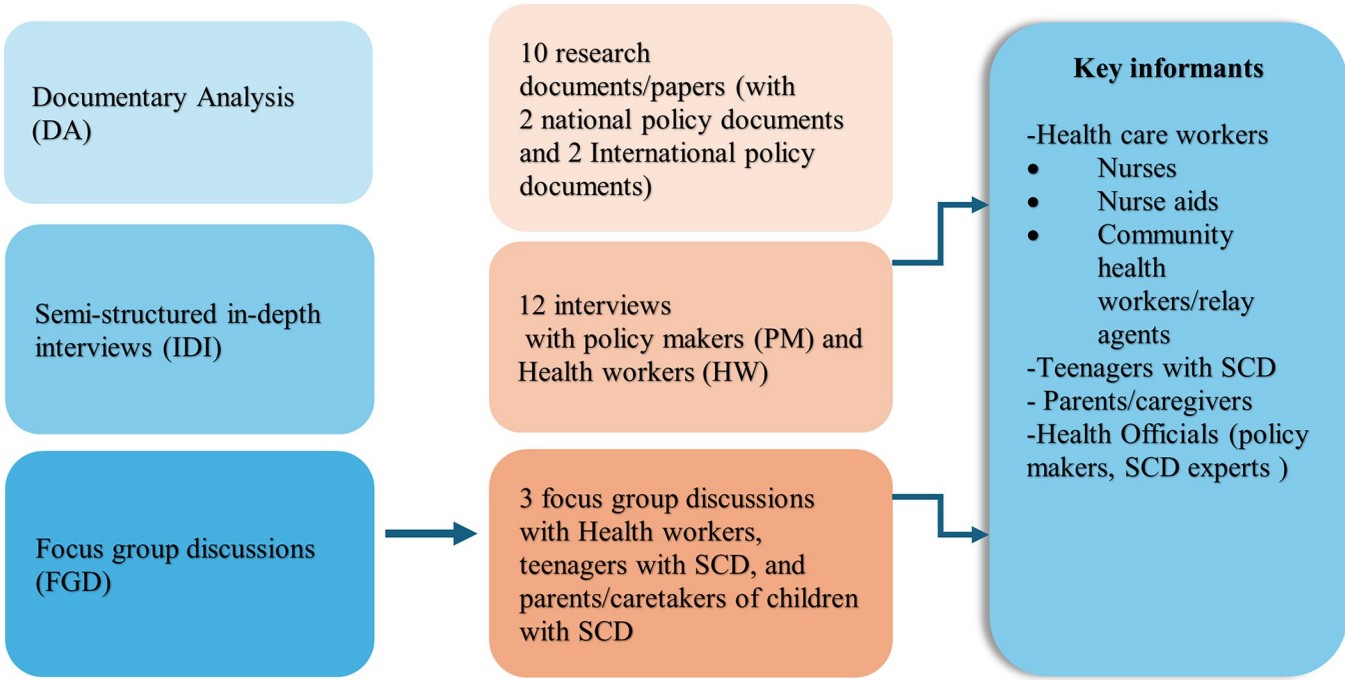

**Fig 1. Data sources and descriptions.**

complete, units of meaning and set of categories were set, and added to the general codes and characteristics later developed for all materials. Interview guides, transcripts and document notes were reviewed based both on a content and thematic analysis frame [19]. An initial thematic frame was developed from this, and used to manually code all material. Main themes were then selected, with cross-checking for consensus and quality amongst team members. A final thematic chart was drawn, and organized retrospectively, with the interpretations of emergent themes presented around recommended health system pillars [20, 21].

## Results and discussion

### The SCD problem in Cameroon: An overview

**1. Screening for prevention and management.** Although genetic testing for SCD or SCT is not a routine component of SCD control in Cameroon, it remains the main stay to prevention against SCD, and the main tool for timely control. Screening for SCD or the SCT can occur at any point throughout life, but is particularly relevant or most valuable during premarital, pre-conceptually, and prenatal periods. in the prenatal period for SCT.

*Premarital screening for SCT*. In Cameroon, premarital (parents or prospective parents) counselling and screening is a preliminary step to church or civil marriage. The goal of this is to allow for the detection of the carrier status of prospective parents, to pre-empt the birth of a child with SCD. This intervention, however, has been reported to have low public health value since most couples decline from changing their marital plans following screening results [22]. Despite this, some respondents in this study felt the role of premarital counseling cannot be completely minimized as quoted below:

> *"Prevention can only be done through pre-marital counselling and screening. . .it is the only recourse for us"* IDI, PM

*"Considering the challenges already in assessing treatment or care for SCD especially in rural areas, the first need then is to avoid the propagation of the disease. Couples must test for SCT before going for children" IDI, HW*

As posited above, lack of access to SCD care and treatment in rural areas in these settings underscore the need for mass awareness via targeted sensitization. Fortunately, some progress has been made in this area in Cameroon. For instance, sensitization messages/information which highlights the need for premarital screening are available and shared in churches, schools, during premarital counselling, and SCD campaigns especially during the international SCD awareness days [23]. Nevertheless, such information, seems to be missed by most population sub-groups, for diverse reasons including inability to read among some population groups, geographical location (especially rural, remote, or urban slums), age, and faith/religion-based beliefs amongst others. Although these factors are known to influence the choices and knowledge on premarital screening, little is known about their potential to influence decisions on the choice of a future life partner, highlighting the needs for further sensitization, as explained by a policy maker:

*". . .this still has to determine peoples' decisions. . .a lot of sensitization is still needed and awareness raising. . .especially with youths" IDI, PM*

This highlights the need to consider intersections around gender, age, education, culture, ethnicity, religion, and geographical location when designing awareness messages for SCT screening. The importance of considering these, have recently been shown in the Cameroonian context and other similar settings, to influence the decision or propensity to accept or present for an SCT screening [22].

Another challenge identified is the lack of tools and systems for data recording in many settings where premarital screening for SCT occurs. Because of this lack, SCT screening data are often lost or become untraceable overtime. This is especially true for community screenings where data are not only isolated but rarely contribute to national SCD statistics. A need for good coordination was highlighted as a necessity in addressing this challenge:

*". . .work is done, the problem is most of the work done is not taken up at the central level or reported here. . .we do not have the data. They come and work and go with their data. We have no idea at times what is going on. There is need for coordination" IDI, PM*

*Prenatal screening for SCT and SCD.* Like premarital screening, prenatal screening for SCT has strong potentials for SCD control. Routine prenatal screening for SCT is retained as part of antenatal care (ANC) in some SCD endemic contexts like Nigeria, Ghana and Burkina Faso. In Cameroon, antenatal consultations have notably increased overtime, and as of 2023, up to "73% of pregnant women attended a prenatal clinic in their first trimester of pregnancy",(DA, [24]). However, screening for SCT during ANC remains unstandardized, a weakness acknowledged by our respondents:

*"it is not certain. . .most areas or hospital, SCT screening is not still prescribed for expecting mothers. . .it has been my experience too" IDI, HW*

The benefits of prenatal SCT screening in pregnant women for an early management of SCD are many-fold, and in most resource limited settings, could replace significantly newborn screening, though the benefits of the latter as well, cannot be underplayed. With the high coverage of ANC in Cameroon, embedding routine and free prenatal screening for SCT could go

a long way to boost prevention and early management of SCD in children, especially in the face of limited uptake of newborn screening for SCD.

In addition to this intervention, early prenatal genetic diagnosis (PND) in unborn babies, is especially recommended before the completion of the first term of pregnancy. This is to enable the identification of SCD in the fetus which in turn can galvanize reproduction options such as a medical termination of the affected pregnancy (TAP). Though some studies show the acceptability of TAP in Cameroon [25, 26], accessing the intervention continues to be a challenge, as highlighted by respondents:

> "PND is rare, and mostly [available] in private health facilities" IDI, HW

> "I don't know where it is done, if at all. . . it most be very few places that do it. I doubt how many couples can afford it" IDI, HW

Asides its availability, other factors such as its invasive nature and cost as well as ethical challenges and lack of legislations for TAP limit its uptake in Cameroon. This later limitation was particularly raised by health workers:

> ". . .abortion is considered illegal under the Cameroonian law, and till date, no amendments have been considered for TAP related to SCD" IDI, HW

> "abortion is permitted if the mother's health is at risk. . .in the specific case of SCD how is this translated? . . . it is an issue" IDI, HW

These findings highlight the need for advocacy and targeted interventions to address ethical, monetary, legal barriers to TAP in Cameroon [12].

*Newborn screening for SCD.* Given the limitations of premarital and prenatal screening, newborn screening (NBS) for SCD remains the most advocated screening tool. Despite this, its uptake remains a challenge in Cameroon and to date data on NBS as well as the distribution of the annual number of children born with SCD are unavailable, except small data sets that come from externally funded projects. In settings with a history of an NBS project, screen activities died out after the project lifespan (and funding).

> ". . .we mostly have no idea what is going on, or by who. Most of the data collected here is exported and without any reporting at our level" IDI, PM

> "the funding ended and so the activities too. The technician at centre Pasteur who was recruited by the project left after. . .nobody updates the data now. It is the problem with such projects" IDI, HW

Despite these limitations, NBS remains the most acceptable (>70%) SCD screening intervention in Cameroon, because of its cheaper costs, less complicated decision making, and lesser stigmatization [22].

**2. Governance and health financing for SCD.** Following the establishment of the Sickle-Cell Disease Strategy for the WHO African Region in 2010, "*[Cameroon], a member state, and the 3ʳᵈ most endemic in the WHO African Region, partially adopted one of the recommended strategies*" DA [9]. Following this adoption, the country established a sub-service incharge of SCD within its Ministry of Public Health. Though this sub-service exists, in practice it is neither centralized nor decentralized, resulting in no clear protocols or directives at the tactical (Regional delegation) and operational (Health District) levels. Thus, SCD management obviously does not come out as a clear priority for the state.

*"...though recommended by the WHO SCD strategy, and verified in 2020, [Cameroon] was not part of the countries to allocate funds in the national budget for surveillance, monitoring and evaluation of SCD activities, nor did she allocate funds for capacity building for prevention and management of SCD, and neither did or has she unlike 7 other high burden countries (Benin, Nigeria, Burkina Faso, Liberia, Mali, Togo and Zambia) allocate funding in her national budget for research related to SCD"DA [27]*

Despite the lack of dedicated funding for SCD in its national budget, the country has made some progress in improving care and treatment for SCD. In 2021, for instance, a detailed care guide for SCD was established for the national territory; however, little is known about the uptake and use of this guideline by healthcare workers and their institutions. Asides clinical guidelines, no other plans, including epidemiological surveillance and financing are detailed in the guide, or exists (DA [28]).

In Cameroon, financing for many conditions is often linked to established programs and services. As such, the lack of a structured program for SCD translates into the lack of a financing plan, implying that state resources are rarely channeled to SCD care and treatment. As a result, the financial burden for SCD related diagnosis, crisis or general management, is born by the patient, and a key challenge described by both SCD patients, their caregivers, and health workers.

*"At times, they could spend all what they had on transportation costs and when they arrived at the hospital, they didn't even have again anything left to pay for consultation or even for the child's treatment..." IDI, HW*

*"...because if the mother has no money, she can't come. Even if she comes with her child and tests and drugs are prescribed, and she is unable to pay for these, the child will not get better...the cost for treatment is a problem for them...for us all. It stops all evolution" FGD, HWs*

*"...the cost for treatment are high and we are feeling guilty to put our families through this. We need financial support or a reduction of treatment costs..."FGD, Teenager with SCD*

These all are a clear technical spill-over from lack of clear programs and strategies at the national level as well as lack of global solidarity for SCD children and their families. Inorder to address these bottlenecks that impede access to SCD care and treatment, there's a need to:

- To establish an autonomous and well-resourced national entity focused on SCD control, with representations in all the regional delegations, health districts and health areas and facilities.

- Develop and execute a clear and costed action plan for SCD, which is aligned to established global policies, strategies, and funding. Key areas to address in the plan should include:

  ○ Prenuptial screening

  ○ Prenatal testing (free routine hemoglobin electrophoresis testing as part of minimum prenatal care package)

  ○ Post-natal management (implement standardized and recorded newborn hemoglobinopathy screening and care)

**3. Health information system for SCD.** Overall, a national system exists for collecting and managing SCD data in Cameroon, though a combination of factors including lack of a data collection policy for sickle cell, lack of data collection tools and indicators to capture SCD

related data on district health information system (DA, [24], were identified as existing gaps. These constitute a gross missed opportunity for SCD control efforts, as it hampers evidence-based decision making and resource allocation, as well as affects the tailoring of messages for sensitization campaigns within communities. This gap was identified as major problem at all levels:

> *"The lack of information routinely collected or existing at any level in the health system is a key gap in the management of SCD in Cameroon"* IDI, PM

In rural areas, access to information (sensitization and communication), was identified as a key obstacle in accessing health care for people with SCD, and reported as the main challenge after finances.

> *"Health services could exist but they [caregivers]are unaware of their existence. . .there is misinformation, disorientation, wrong communication. . .the populations need to be reeducated or informed. they need to understand the realities and existence of SCD as a condition. In these rural areas most people don't know. . .IDI, HW*

To close this gap, there's a need to set up data systems that could systemically collect data on all aspects of SCD. The systems should permit data disaggregation, for example by sex, age, location etc, for more effective and inclusive policies on prevention and management of SCD [29]. This would be necessary for equitable management of the disease, including specific complications which affect daily living for SCD patients [30].

**4. Health workforce (and service delivery) in the management of SCD.** *Health workforce.* The skill set of the health workforce can be constructively guided by a Comprehensive health care management (CHCM) package for SCD. The recommended CHCM for SCD involves new-born screening (NBS), routine follow up, infection prevention and parental education amongst others [4]. Although this package is widely available in high-income countries and in several SSA settings, Cameroon is still to introduce this. Because of this lack, case management for SCD is largely dependent on personal knowledge and practice of physicians.

> *"Given the absence of an established CHCM system in Cameroon, treatment of SCD mostly depends on the initiative of individual health providers"* IDI, HW

This finding questions the quality of care received by persons with SCD, considering many healthcare workers in this setting appear to lack requisite knowledge on SCD care and treatment. In a study assessing the knowledge of various components of SCD care and treatment in over 200 physicians in Cameroon, 5 in 6 (86%) surveyed physicians were found to lack good knowledge about SCD [31]. **Table 1** reports points highlighted by respondents on health worker challenges in their response to SCD.

*Service delivery.* Only a handful of facilities-mostly faith-based, were reported to offer care and treatment services for SCD in Cameroon according to our study respondents.

> *"Care for SCD is available in some private and missionary hospitals like the Cameroon Baptist Convention hospitals. As well, key public hospitals like the Central Hospital in Yaoundé, and the Chantal Biya Foundation (a specialist hospital for HIV and children cancers), offer diagnosis and basic management. I am not sure where else"* IDI, HW

Surprisingly, most public facilities (at secondary and primary levels), which serve most of the Cameroon's population do not offer SCD care and treatment services. This service delivery

**Table 1. Results table to show some strength-weakness linkage commonalities from results.**

| Recommended Action | Existing Action (strengths) | Challenges |
|---|---|---|
| **Routine (newborn) screening** | • Few newborn screening centers (though these are all pilot studies and mostly stop after the project lifespan). Large pilot of 60,000 neonates screening carried out funded by private organization (external funding)<br>• National reference laboratory offers discounted screening or children <15 years old | • No national regulations or program on newborn screening<br>• Poor visibility, and scarcity of reliable, precise and representative data on newborn screening (from current or past pilot NBS studies). Results from pilot studies are not available at national level due to coordination gaps.<br>• Unsustainability of externally funded isolated screening programs (no continuity)<br>• Most screening for SCD (children and adults) comes from clinical suspicion; or pre-nuptial screening. Screening is not routine or standardized<br>• Antenatal testing for SCT and Newborn screening not systematic across health institutions, and services for these paid out of pocket<br>• Limited availability of screening services and equipment, especially in rural areas.<br>• Limited use of point-of-care testing: Traditional methods are resource and time intensive. |
| **Infection prevention, management, and routine follow-up** | • Select SCD clinics exist within specialized tertiary level hospitals. In a few public PHCs and private health facilities (e.g. in the CBCHS (Cameroon Baptist Convention Health Services)), there are sickle cell clinics set up for SCD care and follow-up.<br>• Prescription for Hydroxyurea (HU) existing,<br>• Existence of some level of routine follow up<br>• No SCD management guidelines with set guidelines or regulations for crisis management;<br>• 5 central public institutions (mostly at tertiary and secondly level) in Douala and Yaoundé equipped to manage SCD.<br>• Booster vaccines available | • Concentration of health facilities offering SCD specialized diagnosis and care in urban areas (mostly to tertiary level health institutions, with <5 public health facilities identified with a dedicated unit to offer specific SCD care. Health centers at health area and health district levels mostly lack the technical and human resource capacity to attend to SCD management.<br>• General medical care algorithm for SCD (Vaccinations, Annual eye exam, Screening labs, Nutrition, Control blood pressure) not established and most follow-up not standard, but inconsistent and practitioner specific<br>• No routine data collected on follow-up for SCD to inform central level and inform policy for SCD. In key sickle cell centers contacted, only paper-based registries are used, with no formal filing system to preserve information<br>• Though HU is prescribed for SCD management, challenges exist such as stockouts, cost fluctuations (up to 20 USD for a single dose), practitioner challenges in prescription, difficulty in child dosing due to capsule formulation<br>• Transition clinics (pediatric to adult) from child to adolescent care inexistent, increasing loss to follow-up for teens<br>• Cost for follow-up/crisis management not subvented causing out-of-pocket-expenditure for general medical care and crisis management, and treatment dropouts<br>• Low vaccine booster uptake for children with SCD (pneumococcal and meningococcal vaccines). This is linked to high costs which limit access. Similarly, for severe pain medication (e.g. Morphine) or general tests/treatment. The cost affects uptake amongst SCD children/adults.<br>• Frequent stockouts of necessary/essential medication. Blood transfusion services are not very available, and not affordable, especially in emergencies.<br>• Limited availability of psychosocial care for adolescents and adults living with SCD. |
| **Community management** | • Some private SCD groups organized and animated, but not monitored or supported by state;<br>• Existing social support in some areas organized by independent NGOs (e.g. CERAC Women donating Iron tablets and animating SCD support groups in Mfou Health District). Such initiatives could be integrated and extended nation wide | • Limited sensitization and thus limited awareness of SCD in most rural areas<br>• Existing SCD support groups unintegrated or unmonitored and at times die-out |

*(Continued)*

**Table 1.** (Continued)

| Recommended Action | Existing Action (strengths) | Challenges |
|---|---|---|
| Health worker capacity | • Availability and willingness of community health care workers to refer SCD and acts as relay in SCD campaigns (isolated study in Mfou Health District (FAHEDEF activity report, 2023))- Possibility of integration within CDD/CHW programs for MDA campaigns for NTDs and EPIs<br>• Existing examples of health practitioners trained and with good knowledge on SCD from benefiting from some CSO organized trainings (e.g. FAHEDEF (Family Health and Development Foundation (civil society organization))) | • Inadequate knowledge and capacity of most frontline health care workers for SCD management; No existing training or capacity building opportunities for frontline staff on SCD management; Limited number of specialists (clinical hematologists/pediatricians working on SCD) in Cameroon (SCD mostly managed by biologists) Poor knowledge of primary care physicians on use of HU<br>• Sustainability of pilot projects involving health care workers (formal and informal) after project lifespan. Skills gained from such not recycled or used optimally without SCD activities constant or followed by State. |

challenge (amongst others highlighted in **Table 1**) invariably contributes to untold pain and suffering for the patients and their families because of their inability to access care and treatment services in a timely manner. This a challenge which has a negative impact on their physical and mental wellbeing. Key service delivery barriers reported by our respondents included:

- Limited skilled professionals on SCD management

- Understaffed health institutions

- Limited infrastructure and technical capacity for SCD management, including screening, diagnosis, specialized care with standard equipment.

- Unstandardized treatment guidelines for SCD (with key gaps in transitioning from pediatric to adult care)

- Treatment challenges driven by unavailability and high cost of Morphine, Hydroxyurea, blood products and other preventative tools such as conjugate vaccines. This challenge is further compounded by prescription challenges amongst medical practitioners causing more side effects, leading to adherence issues.

- High treatment cost for SCD, which is unsubsidized and born by the family

From the patient angle, care givers of SCD patients highlighted other factors limiting access to care services even when they are available. These include lack of timely access to key information, ignorance of SCD as a medical condition, mishandling of caregivers and patients by health personnel, high cost of health care, distance to SCD clinics for follow-up and stigma faced by the patients, amongst others. In a focus group with SCD teenagers and caregivers, teenagers with SCD especially highlighted mockery at school or lack of general support at hospital or within the community; and their desire to be accepted by the society as normal children, as their biggest challenges.

> When I *am at school, the others [kids] laugh at me. . .they say I have a big head. As the veins usually come out also, they laugh that I have spaghetti veins. . .Female, 13 years old*

> *We just want to be loved, treated like normal children. . .Male, 15years old*

> "*When they see a youth, they think you are pretending and seeking attention. It is different from when you are younger; also lack of support or stigmatization from the community*: *insults in school. . .Female, 16years old*

> "*At the hospital they should respect us and treat us like every other patient, and not make us feel rejected. . .Male 15years old*

Similar observations have been reported from settings, which also appear to stem from ignorance [6, 8]. Fortunately, these challenges can be attenuated with heightened awareness raising and education on SCD [32] in schools and communities. These awareness campaigns should equally target healthcare workers, who also face significant challenges when it comes to SCD management. Ideally, the awareness campaigns should focus on mobilizing the necessary resources to build the capacity of health care workers on addressing stigmatization in hospitals, schools, and communities, advocating for pre-marital screening and dispensing quality care to SCD patients. The later should include provision of screening, prophylaxis against infection, acute medical care, safe blood transfusion, and hydroxyurea (HU) in line with standard recommendations [4].

**5. Access to essential medicines.** Like in many resources poor settings, Hydroxyurea (HU) remains the most prescribed diseases modifying agent in Cameroon, with its value highlighted by respondents as the key reason for this.

*". . .it has. . .reduced emergency hospital visits, hospitalizations, even need for blood transfusions and other pain management, when the patient is put on it for a while. . .but this is when the dosage is given right by the physician" IDI, HW*

However, its use is marred by multiple challenges, including availability, cost, and prescription by practitioners. Although its cost was described by informants at the central level as "relatively affordable" (less than 1,500USD for two years treatment for young children), its cost remains an economic burden which cannot be contested [33]. The current cost appears to be relatively high for an average Cameroonian household with per capita household disposable income estimated at 2.76USD [34], re-affirmed by respondents at the operational level:

*"the cost of the treatment is the problem. . . the government needs to help" FGD, HWs*

*"what we need is money for our family to buy our medicines. . . and that the medicine should be available too" FCD, Teenager with SCD*

As highlighted here, the availability of HU in the local market is a challenge. This is partly driven by the limited purchasing power of families, which apparently do not offer sufficient incentives for distributors and pharmacies to import and commercialize the medicine in Cameroon. This situation often results in stockouts, affecting those most in need. The situation also creates an ethical dilemma with regards to health equity, highlighting an urgent need in subsidies. Without such an intervention, many SCD patients and families may be unable to bear the high treatment cost as reported elsewhere [35, 36].

Another fundamental challenge highlighted in the use of HU was the inability of practitioners to prescribe the right doses to patients. This challenge increases the likelihood of toxicity, which it was reported to affect adherence and ultimately leading to suspension of treatment by patients or a complete rejection. As a result, many clinicians were found to refrain from prescribing the drug as posited below:

*"it is tough to prescribe. . .most doctors avoid prescribing it because of dosage uncertainties. . ."IDI, HW*

*At times, without the right dosage, it could cause severe after-effects, and this this one of the main reasons why SCD patients refuse taking it too. . .the myths around HU like infertility, etc are mostly as a result of wrong dosage. . ."IDI, HW*

In addition to the identified challenges related to HU access and use, the scarcity of blood banks, limited access to pain medications such as morphine, and antibiotics were all raised as current challenges in relation to treatment/management of SCD by health workers and the persons affected by SCD.

Recently, phenomenal strides have been made to develop new treatments for SCD [4]. Although curative treatment (gene editing) is now possible, these alternatives are only available to a tiny fraction of patients in high-income countries because of the prohibitive cost [4]. Additionally, other novel products such as Crizanlizumab, Voxelotor, and Lglutamine have been recently approved, but their availability would remain a distant goal for SCD patients in Cameroon [32]. However, accessing these revolutionary therapies may be a game changer on the burden of SCD in Cameroon, though the direct cost and indirect costs such as productivity loss and missed work for caregivers or SCD patients from pain episodes remain unmeasured.

## Recommendations for policy and integration

Based on the above analysis, we propose recommendations (Table 2) to improve SCD management in Cameroon grouped around 3 strategic points: strengthening governance through the functionality of a national SCD strategy; Coordinating and integrating services within existing

**Table 2. Opportunities to leverage for sustainable integration.**

| Domain | Key Activities |
|---|---|
| **Strengthening policy and governance frameworks.** | • Establish a national SCD program and strategies, with attached regional, health district and health area structures and focal points<br>• Attribute proportional national financing for the SCD program and aligned activities (subsidies for essential medicines, research and innovation), and promote integration within health delivery programs especially at the intermediary and peripheral/operational levels.<br>• Implement national advocacy and impel the State to create and allocate funding from national budget (and management structures) for subsidizing or totally covering early diagnosis, management and surveillance for SCD across the different age groups;<br>• Establish a clear mechanism to monitor and report age-aggregated and sex-aggregated data on SCD (information on births, diagnosis, complications and SCD related deaths) for effective management, with clear and equitable programs and policies for SCD prevention and control at all levels.<br>• Create a platform to harmonize all research on SCD in the country to inform policy and limit over-laps of research and financing for programs. |
| **Coordinating and integrating services** | • Facilitate the integration of SCD care and treatment into primary care and existing programs at all levels of the health system, providing a unique opportunity to minimize cost and maximize synergies for greater transformational impact. Such as:<br>○ Vaccine Delivery: Incorporate newborn screening into existing routine immunization program as seen in a pilot point of care testing integration into routine immunization in Nigeria [37]. Strengthen immunization programs to provide Life Course Vaccination for Children and Adults with SCD.<br>○ Prenatal care: Establish early detection and management of SCD by leverage antenatal care within intermediate and peripheral health level facilities to establish prenatal counselling and routine prenatal screening to identify mothers at higher risk of giving birth to children living with SCD and enabling prompt diagnosis and follow up.<br>○ Build on existing infectious disease programs like HIV and malaria care systems to expand SCD screening and care as seen in Uganda [38].<br>○ Community Health Programs: Integrate SCD prevention and care training into existing community health worker (CWH/CDD) programs, leveraging established recruitment, training, and deployment systems. Existing mass drug administration (MDA) programs could be leveraged for SCD sensitization and integrated distribution of folic acid for adults and children living with SCD to maximize cost-effectiveness.<br>○ Improve access to essential medication for SCD by integrating essential pain medication (e.g., opioids) and HU into established procurement and distribution channels for cancer medicines.<br>• Coordinate the management of individuals living with SCD of all ages, ensuring continuous and subsidized availability of HU and other essential medication for SCD care in health facilities at all levels of health care delivery, especially at primary health care facilities. This is more common at tertiary health facilities, and a few secondary health facilities, placing rural populations at a detriment. |

*(Continued)*

**Table 2.** (Continued)

| Domain | Key Activities |
|---|---|
| **Creating an enabling environment by engaging and empowering communities.** | • Train community health workers and primary health level health workers on SCD prevention, care, and treatment at the health district and health area levels.<br>• Create and accompany community SCD advocacy and support groups within communities at all levels.<br>• Raise awareness for SCT screening benefits within communities especially within primary health care, and educate on SCD to dispel myths and combat stigma and discrimination to support mental wellness and psychosocial health of people living with SCD and their immediate entourage |

health system services and programs; and creating an enabling environment by empowering and engaging communities.

This study did not fully consider the perceptions of other population groups of people with SCD other than teenagers, leaving out substantial information, as well as did not collect or consider (other than within the literature) quantitative data on the state, control and knowledge of SCD control in Cameroon. Also, the document analysis carried out, covered very few documents, but which only go to attest on the limited existing policy documents and reports on SCD in Cameroon.

## Conclusion

The specificity of SCD in Cameroon is intricately linked to the existent high burden and the alarming state of "unmangement" or missed opportunities for control. Several prerequisites need to exist for a favorable environment to control and manage SCD in Cameroon. Firstly, an awareness and understanding of SCD within the general population is certain to encourage more premarital tests for hemoglobin genotyping, diagnosis and follow-up of treatment for children, and adults with SCD. This, in an enabling environment with sound programs and strategies, with required funds allocated for these strategies, will improve the healthcare of SCD patients, and also, alleviate stigma from ignorance. Furthermore, the training and knowledge of SCD among healthcare professionals is a key function, for improved health service delivered for people living with SCD. In addition, public financing, will facilitate availability, affordability and over all access to systematic infant screening, availability of standard health services and drugs for pain and related complications management. As well, precise surveyance and collection of representative epidemiological data is key in informing and directing development of clear and sufficient public health policies and funding allocation. These all translate into clear decision making and planning on SCD management.

## Supporting information

**S1 Table. List of documents collected for documentary analysis.**
(DOCX)

## Acknowledgments

The authors would like to acknowledge Paul Domanico, Elizabeth McCarthy, and Sheetal Ghelani for their guidance in developing the data collection framework. The authors also acknowledge all the participants/informants of this study for their selfless contribution in sharing their experiences, challenges, and strengths in working on, or living with sickle cell disease as a patient or the caretaker of a child or teenager with SCD.

## Author Contributions

**Conceptualization:** Yauba Saidu, Elvis Ndansi, Munoh Kenne Foma.

**Data curation:** Yauba Saidu, Makia Christine Masong, Nwabufo Francoise, Budzi Michael Ngenge, Elvis Ndansi, Munoh Kenne Foma.

**Formal analysis:** Makia Christine Masong, Budzi Michael Ngenge, Elvis Ndansi, Munoh Kenne Foma.

**Funding acquisition:** Yauba Saidu.

**Investigation:** Nwabufo Francoise, Budzi Michael Ngenge.

**Methodology:** Yauba Saidu, Makia Christine Masong, Budzi Michael Ngenge.

**Project administration:** Yauba Saidu.

**Resources:** Yauba Saidu.

**Visualization:** Makia Christine Masong.

**Writing – original draft:** Yauba Saidu, Makia Christine Masong, Munoh Kenne Foma.

**Writing – review & editing:** Yauba Saidu, Makia Christine Masong, Nwabufo Francoise, Budzi Michael Ngenge, Elvis Ndansi, Munoh Kenne Foma.

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
