## [Decision Letter · Decision Letter 0]

7 Jun 2024

PGPH-D-24-00845

Sickle Cell Disease in Cameroon: Taking out the “neglect” and highlighting key opportunities for sustainable control

Dear Dr. Saidu,

Thank you for submitting your manuscript to PLOS Global Public Health. After careful consideration, we feel that it has merit but does not fully meet PLOS Global Public Health’s publication criteria as it currently stands. Therefore, we invite you to submit a revised version of the manuscript that addresses the points raised during the review process.

The authors have done a good study on a  serious genetic disorder with a public health importance especially on access to care and barriers.The results become more important as the key informants includes the policy makers and the patients themselves and the parents.There are critical gaps were identified which highlights the need for policy decisionsHowever , there are certain changes required which will  make tthe impact of his manuscript more perceivable.

We look forward to receiving your revised manuscript.

Kind regards,

Suma Krishnasastry, MBBS, MD,DNB, FRCP (Edin)

Academic Editor

Journal Requirements:

Additional Editor Comments (if provided):

Reviewers' comments:

Reviewer's Responses to Questions

**Comments to the Author**

1. Does this manuscript meet PLOS Global Public Health’s publication criteria? Is the manuscript technically sound, and do the data support the conclusions? The manuscript must describe methodologically and ethically rigorous research with conclusions that are appropriately drawn based on the data presented.

Reviewer #1: Yes

Reviewer #2: Yes

2. Has the statistical analysis been performed appropriately and rigorously?

Reviewer #1: Yes

Reviewer #2: N/A

3. Have the authors made all data underlying the findings in their manuscript fully available (please refer to the Data Availability Statement at the start of the manuscript PDF file)?

Reviewer #1: Yes

Reviewer #2: Yes

4. Is the manuscript presented in an intelligible fashion and written in standard English?

Reviewer #1: Yes

Reviewer #2: Yes

5. Review Comments to the Author

Reviewer #1: The manuscript presents an insightful and comprehensive qualitative study exploring the barriers to screening, care, and treatment of Sickle Cell Disease (SCD) in Cameroon. By engaging multiple stakeholders, including policy makers, healthcare workers, parents, and teenagers living with SCD, the study highlights critical gaps. The research is timely and addresses a significant public health issue, providing valuable contributions to the field. However,

1. The description of the study design and data collection methods lacks sufficient detail.

2. There are inconsistencies in terminology, such as switching between "SCD" and "SCT" without clear definitions or distinctions.

3. The recommendations are broad and lack specific implementation strategies.

Reviewer #2: Good study that could assess the current scenario of SCD and highlight the need for policy level actions. If approximately 7000 children are born with SCD annually in the country, more quantitative data analysis would definitely be helpful.

6. PLOS authors have the option to publish the peer review history of their article (what does this mean?). If published, this will include your full peer review and any attached files.

**Do you want your identity to be public for this peer review?** For information about this choice, including consent withdrawal, please see our Privacy Policy.

Reviewer #1: No

Reviewer #2: No

---

## [Editor Report · Decision Letter 1]

15 Aug 2024

Sickle Cell Disease in Cameroon: Taking out the “neglect” and highlighting key opportunities for sustainable control

PGPH-D-24-00845R1

Dear Dr Saidu,

We are pleased to inform you that your manuscript 'Sickle Cell Disease in Cameroon: Taking out the “neglect” and highlighting key opportunities for sustainable control' has been provisionally accepted for publication in PLOS Global Public Health.

Best regards,

Suma Krishnasastry, MBBS, MD,DNB, FRCP (Edin)

Academic Editor